anxiety; depression; mental disorders; Nepal; global burden of disease

**Corresponding author:**
Raja Ram Dhungana;
Email: raja.dhungana@gmail.com

# The burden of mental disorders in Nepal between 1990 and 2019: Findings from the Global Burden of Disease Study 2019

Raja Ram Dhungana[1,2] [ID], Achyut Raj Pandey[3] [ID], Suira Joshi[4],
Nagendra P. Luitel[5] [ID], Kedar Marahatta[6], Krishna Kumar Aryal[7] [ID] and
Meghnath Dhimal[8] [ID]

[1]Manmohan Memorial Institute of Health Sciences, Kathmandu, Nepal; [2]Faculty of Medicine, Nursing and Health Sciences, Monash University, Australia; [3]HERD International, Kathmandu, Nepal; [4]Ministry of Health and Population, Kathmandu, Nepal; [5]Transcultural Psychosocial Organization Nepal, Kathmandu, Nepal; [6]World Health Organization, Country Office for Nepal, Kathmandu, Nepal; [7]Bergen Centre for Ethics and Priority Setting in Health, Department of Global Public Health and Primary Care, University of Bergen, Bergen, Norway and [8]Nepal Health Research Council, Kathmandu Nepal

## Abstract

Mental disorders are the leading cause of disease burden, affecting 13% of all people globally in 2019. However, there is scarce evidence on the burden of mental disorders in Nepal. This study used the Global Burden of Disease Study 2019 data to assess the prevalence and disability-adjusted life-years (DALYs) of mental disorders in Nepal between 1990 and 2019. In 2019, there were 3.9 million (95% UI: 3.6–4.3) people with mental disorders in Nepal. Major depressive disorders (1.1 million; 95% UI: 0.9–1.2 million) and anxiety disorders (0.9 million; 95% UI: 0.8–1.2 million) were the most prevalent mental disorders in 2019. Attention deficit hyperactive disorder, conduct disorder, and autism spectrum disorders were present twice as high in males than in females. The proportional contribution of mental disorders to the total disease burden has tripled between 1990 (1.79% of all DALYs) and 2019 (5.5% of all DALYs). In conclusion, the proportional contribution of mental disorders to total disease burden has increased significantly in the last three decades in Nepal, with apparent sex and age differentials in prevalence and DALY rates. Effective program and policy responses are required to prepare the health system for reducing the growing burden of mental health disorders in Nepal.

## Impact statement

Nepal, having endured a decade-long internal armed conflict, and grappling with the aftermath of devastating events such as the 2015 earthquake, and other multitude of personal, social, cultural, economic, political, and environmental adversities, lacks a systematic analysis of the trend and burden of mental disorders. This study fills an important gap by providing estimates of the prevalence and disability-adjusted life-years (DALYs) associated with mental disorders in Nepal between 1990 and 2019. The study emphasizes the increasing burden of mental disorders in Nepal over the past three decades, with notable sex and age differences in prevalence and DALY rates. The findings indicate that in 2019, Nepal had approximately 3.9 million people with mental disorders, where major depressive disorders and anxiety disorders were the most prevalent conditions. The proportional contribution of mental disorders to the total disease burden has tripled between 1990 and 2019. The significant findings indicate a need for effective program and policy responses to address the growing burden of mental health disorders in Nepal. These findings may also inform the stakeholders for preparing the health system to meet the challenges posed by mental disorders.

## Introduction

Mental disorders are the major cause of disease burden globally. The proportional contribution of mental disorders to total disability-adjusted life-years (DALYs) has increased by 58% between 1990 and 2019 globally (Institute of Health Metrics and Evaluation, 2019). In 2019, mental disorders were the seventh leading cause of DALYs, affecting 970 million people around the world (Institute of Health Metrics and Evaluation, 2019). Mental disorders are also responsible for a high economic burden resulting from increased healthcare expenditure and productivity losses (Doran and Kinchin, 2019). Low-income and middle-income countries (LMICs) like Nepal are disproportionately affected by the burden of mental disorders. In 2019, most of the

people with mental disorders were from LMICs (Institute of Health Metrics and Evaluation, 2019).

Exposure to several personal, social, cultural, economic, political, and environmental adversities including chronic health conditions, impoverishment, social exclusion, gender disadvantage, conflict, disasters, and migration, among others, could determine the development of mental disorders (Lund et al., 2018). Passing through the decade-long internal armed conflict during 1996–2006 (Medeiros et al., 2020), experiencing massive disasters like the earthquake in 2015 (Kane et al., 2018), high unemployment rate, a substantial rate of out-migration (Dhungana et al., 2019), high incidence of domestic violence, high alcohol consumption rates, and the pernicious problem of poverty could be predisposing factors for poor mental health in Nepal (Luitel et al., 2013).

A recent national mental health survey reported that 10% of Nepalese adults had any mental disorder in their lifetime (Nepal Health Research Council, 2021). Other studies also provided estimates of particular mental disorders such as anxiety and depression in Nepal (Upadhyaya and Pol, 2003; Khattri and Nepal, 2006; Kohrt et al., 2009; Risal et al., 2016; Simkhada et al., 2018). However, these studies were conducted in specific samples (Clarke et al., 2014; Dhungana et al., 2019) or used different self-reported assessment tools with limited validity (Upadhyaya and Pol, 2003; Khattri and Nepal, 2006; Kohrt et al., 2009; Risal et al., 2016), which might have contributed to a wide variation in the reported rates of mental health problems in Nepal (Steel et al., 2009). Most importantly, neither of the previous studies attempted to estimate DALYs nor assessed the temporal trends in the prevalence of mental disorders in Nepal.

Moving beyond the studies whatsoever available that provide a descriptive picture on the prevalence of selected mental disorders in Nepal, and within the context of poor availability of and access to mental health care, a systematic analysis of the trend and burden of mental disorders can inform stakeholders about the magnitude and distribution of comprehensive measures of the burden of mental disorders in Nepal. A clear understanding of the extent of mental health problems in the population is crucial for planning and implementing effective prevention and management strategies in Nepal. Therefore, this study aimed to illustrate the trend and pattern of mental disorders in terms of their prevalence and DALYs from 1990 to 2019 using the data from the Global Burden of Disease Study (GBD) 2019.

## Methods

### Study design and data sources

This study was based on the estimates provided by GBD 2019. We extracted the data from the official website (http://ghdx.healthdata.org/gbd-results-tool) of the Institute for Health Metrics and Evaluation (IHME) using the 'GBD Compare' data visualization tool (Institute of Health Metrics and Evaulation, 2020). GBD study collects a wide range of data from various sources, including vital registration systems, health surveys, disease registries, healthcare facilities, and more and analyzes those data using sophisticated statistical techniques and modeling to estimate the incidence and prevalence, deaths, and DALYs attributed to specific diseases. The GBD 2019 was a multinational collaborative study that covered 204 countries and regions and provided a comprehensive assessment of health loss for 369 diseases and injuries from 1990 to 2019 (Roth et al., 2020). The GBD 2019 used a total of 281,577 data sources globally and 402 data sources from Nepal to estimate the

disease burden. The data input sources comprised household survey data, hospital administrative data, and disease registries, among others.

### Study outcomes

Outcome variables comprised of a list of mental disorders including major depressive disorders (major depressive disorder and dysthymia), anxiety disorders (a combined estimate of all subtypes), idiopathic developmental intellectual disability (estimated within the wider scope of intellectual disability impairment, encompassing cases of intellectual disability originating from unidentified sources once all other potential causes have been considered), dysthymia, attention deficit hyperactivity disorder, conduct disorder, bipolar disorder (a combined estimate of all subtypes), autism spectrum disorders, schizophrenia, bulimia nervosa, anorexia nervosa, and other mental disorders. The mental disorders were defined based on the Diagnostic and Statistical Manual of Mental Disorders or the International Classification of Diseases-10 criteria (Vos et al., 2020).

### Data analysis

A detailed description of the statistical modeling for mental disorders has been reported elsewhere (Vos et al., 2020). In brief, Years Lived with Disability (YLDs) were calculated by multiplying prevalence estimates across different degrees of severity by a relevant disability weight. These disability weights were used to assess the level of health loss associated with each subsequent consequence (due to an illness or accident). Years of Life Lost (YLLs) were calculated by multiplying the number of deaths attributed to a specific cause by the predicted remaining years of life at the time of death, as determined by a standard life expectancy measurement. The combined sum of YLDs and YLLs was used to calculate DALYs. In cases where mental diseases were not officially recognized as direct causes of mortality, YLL computations were removed, and YLDs were employed as an approximation for DALYs.

We used the overall and sex-specific crude and age-standardized rates and the 95% uncertainty interval (UI) from 1990 to 2019 to compare and depict the trends of prevalence and DALYs associated with each mental disorder. We also plotted line graphs of every mental disorder across the age groups ranging from less than 1 year to above 80 years. STATA software version 16.1 (Stata Corporation, College Station, TX, USA) was used to construct the graphs. The prevalence and DALY rates were presented per 100,000 population.

## Results

In 2019, 3.9 million (95% UI: 3.6–4.3) Nepalese were estimated to have suffered from mental disorders, comprising 13.5% of the total population. The overall prevalence of mental disorders in 1990 was 12.4% (95%UI: 10.9–13.8). The prevalence rates did not differ significantly between 1990 and 2019 (Figure 1).

### Prevalence of mental disorders

The age-standardized prevalence of mental disorders per 100,000 population was 13,372.2 (95% UI: 12,144.4, 14,565.75) in 2019 and 13,819.75 (12,444.59, 15,140.57) in 1990, respectively. In 2019, age-standardized prevalence cases of mental disorders per 100,000 population among males and females were 13,023.79 (95% UI:

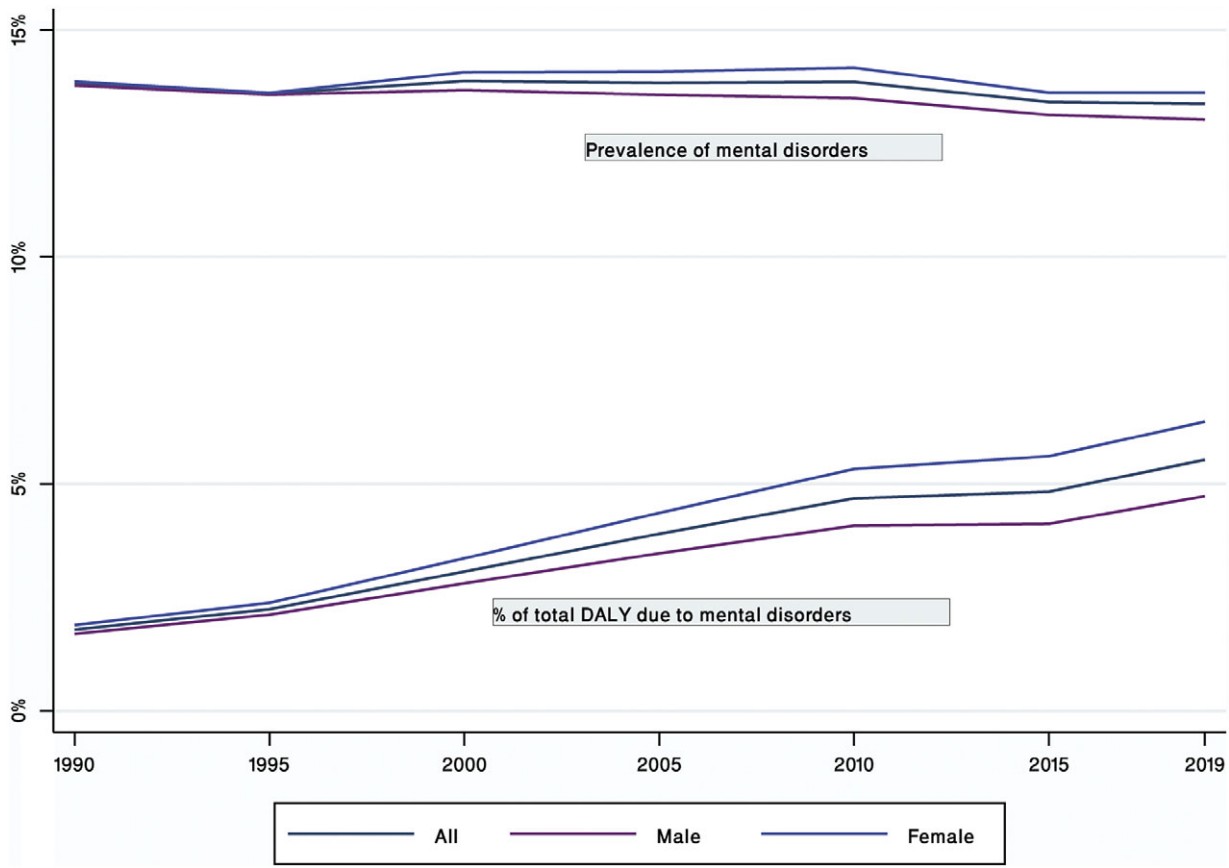

**Figure 1.** Trends in the prevalence of mental disorders and proportional contribution to overall DALYs between 1990 and 2019.

**Table 1.** Prevalence of mental disorders (per 100,000)

| | Both | | Male | | Female | |
|---|---|---|---|---|---|---|
| | All ages | Age-standardized | All ages | Age-standardized | All ages | Age-standardized |
| **1990** | 12,223.2 | 13,819.75 | 12,426.19 | 13,776.17 | 12,019.18 | 13,861.59 |
| | (10,782.5, 13,571.14) | (12,444.59, 15,140.57) | (10,956.48, 13,844.77) | (12,340.34, 15,155.77) | (10,639.14, 13,387.3) | (12,508.23, 15,214.98) |
| **1995** | 12,064.11 | 13,593.67 | 12,283.4 | 13,571.47 | 11,844.05 | 13,610.84 |
| | (10,714.06, 13,366.86) | (12,299.91, 14,912.97) | (10,864.85, 13,643.94) | (12,204.15, 14,895.13) | (10,553.44, 13,169.67) | (12,329.9, 14,974.64) |
| **2000** | 12,377.25 | 13,871.96 | 12,444.68 | 13,673.15 | 12,309.65 | 14,068.18 |
| | (11,039.7, 13,637.37) | (12,621.08, 15,102.55) | (11,045.78, 13,777.93) | (12,359.4, 14,981.86) | (11,042.08, 13,634.05) | (12,732.1, 15,396) |
| **2005** | 12,580.71 | 13,831.49 | 12,539.22 | 13,568.92 | 12,621.44 | 14,080.45 |
| | (11,252.63, 13,853.11) | (12,585.77, 15,095.36) | (11,155.82, 13,840.08) | (12,256.05, 14,871.02) | (11,380.77, 13,932.44) | (12,818.85, 15,395.19) |
| **2010** | 12,873.25 | 13,854.38 | 12,653.49 | 13,494.49 | 13,082.27 | 14,165.59 |
| | (11,652.61, 14,077.42) | (12,652.53, 15,003.48) | (11,342.78, 13,906.57) | (12,196.29, 14,739.67) | (11,847.32, 14,310.12) | (12,872.33, 15,358.35) |
| **2015** | 12,814.23 | 13,410.21 | 12,545.45 | 13,126.49 | 13,063.05 | 13,618.11 |
| | (11,596.87, 14,011.1) | (12,186.09, 14,596.8) | (11,299.25, 13,794.76) | (11,875.24, 14,343.81) | (11,774, 14,410.75) | (12,325.36, 14,918.14) |
| **2019** | 13,003.36 | 13,372.2 | 12,586.64 | 13,023.79 | 13,382.89 | 13,617.99 |
| | (11,765.55, 14,198.43) | (12,144.4, 14,565.75) | (11,315.23, 13,831.36) | (11,777.42, 14,245.63) | (12,118.48, 14,685.06) | (12,372.86, 14,886.33) |

11777.42, 14,245.63) and 13,617.99 (95% UI: 12372.86, 14,886.33), respectively (Table 1).

Major depressive disorders (age-standardized prevalence rate: 3,795.9, 95% UI: 3,265.67, 4,408.13), anxiety disorder (age-

standardized prevalence rate: 3,277.32, 95% UI: 2,624.16, 4,134.75), and idiopathic developmental intellectual disability (age-standardized prevalence rate: 2,503.69, 95% UI: 1,635.85, 3,380.81) were the three most prevalent mental disorders in 2019 (Figure 2).

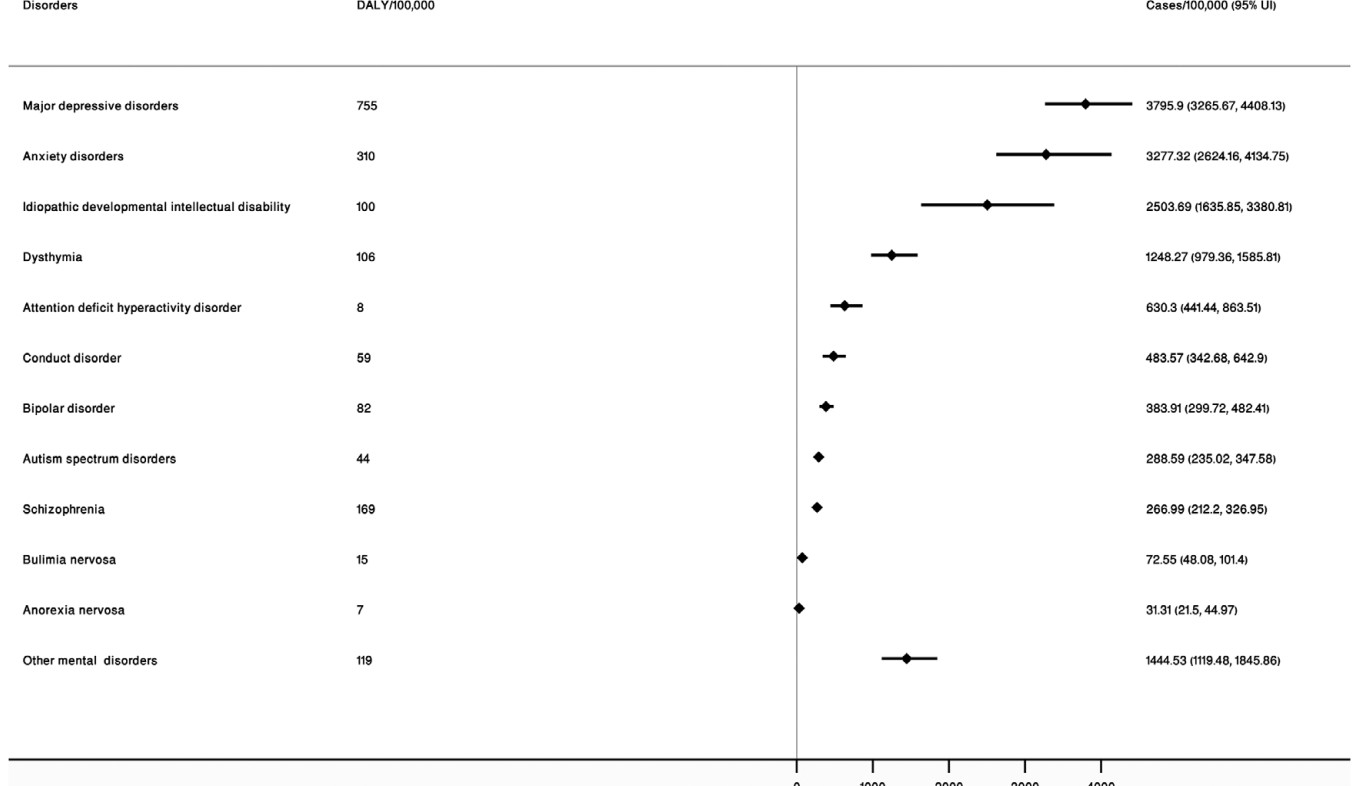

**Figure 2.** Prevalence of and DALYs due to mental disorders in 2019.

Major depressive disorders and anxiety disorders were more common in females than in males (Supplementary Table S1). Likewise, attention deficit hyperactivity disorder, conduct disorder, and autism spectrum disorders were more prevalent in children and adolescents (Figure 3).

### Burden of mental disorders

All age DALYs per 100,000 population increased from 1,421.66 (95% UI: 1,036.74, 1,870.04) to 1,691.08 (95% UI: 1,244.47, 2,224.51) in both sexes; 1,375.63 (95% UI: 997.36, 1,803.81) to 1,545.37 (95% UI: 1,129.01, 2,027.12) in males; and 1,467.92 (95% UI: 1,068.11, 1,944.53) to 1,823.78 (95% UI: 1,330.85, 2,413) in females between 1990 and 2019 (Supplementary Table S2). The proportion of DALYs attributable to mental disorders has increased significantly from 1990 to 2019 (Figure 1) from 1.79% of total DALYs (95% UI: 1.34, 2.32) to 5.53% of total DALYs (95% UI: 4.22, 6.98) in both sexes; 1.7% of total DALYs (95% UI: 1.26, 2.21) to 4.73% of total DALYs (95% UI: 3.58, 6) in males; and 1.89% of total DALYs (95% UI: 1.41, 2.44) to 6.37% of total DALYs (4.89, 7.96) in females (Supplementary Table S2).

The major share of DALYs attributable to mental disorders was due to major depressive disorders and anxiety disorders (Figure 2). All age and age-standardized DALYs for major depressive disorder were 687.86 (95% UI: 465.55, 953.1) and 754.62 (95% UI: 510.96, 1,045.8) per 100,000 in 2019, respectively. Disaggregated by sex, both the overall and age-standardized DALY rates were higher in females compared to males for major depressive disorders. Age-standardized DALYs for major depressive disorders for 100,00 males and females were 605.29 (95% UI: 403.62, 842.67) and 883.21 (95% UI: 598.08, 1,228.91), respectively. Similarly, age-

standardized DALYs for anxiety disorders was 309.51 (95% UI: 205.76, 434.63) per 100,000 in both sexes (Supplementary Table S3).

### Discussion

We found the proportional contribution of mental disorders to the total disease burden has tripled between 1990 and 2019 in Nepal. Major depressive disorders and anxiety disorders, which were also more prevalent in females than males, were the top two contributors to the total prevalent cases and DALYs in 2019. Attention deficit hyperactive disorder, conduct disorder, and autism spectrum disorders were present twice as high in males than in females.

Our study found that one among seven Nepalese had a mental disorder in 2019. The rate is slightly greater than the prevalence of any mental disorder reported in the first national Mental Health survey-2020 in Nepal (Nepal Health Research Council, 2021) and consistent with the prevalence reported in the study (based on GBD estimates) from a neighboring country, India (14.3%) (Sagar et al., 2020). Comparing the overall prevalence of mental disorders within and across countries is still challenging due to inconsistencies in defining and estimating the burden of mental disorders (Kohrt and Hruschka, 2010; Whiteford et al., 2016). For example, GBD 2019 classified mental disorders, neurological disorders, and suicide separately, while other studies grouped them to estimate the burden of mental disorders (World Health Organization, 2018; Rehm and Shield, 2019). The national Mental Health Survey also included substance use disorders to report the prevalence of any mental disorder in Nepal (Nepal Health Research Council, 2021). Regarding the major depressive disorders, most of the South Asian

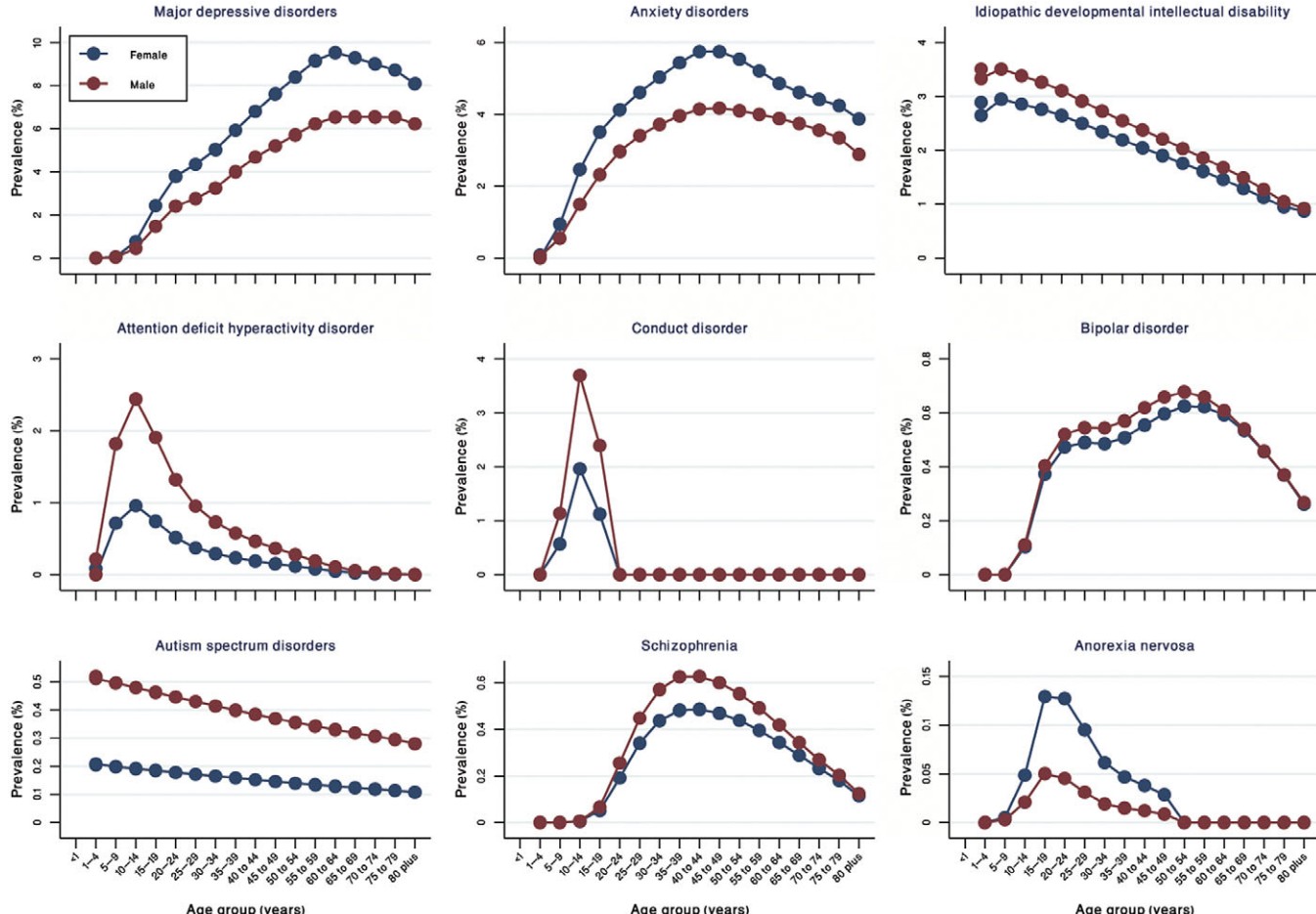

**Figure 3.** Prevalence of mental disorders by age and sex in 2019.

counterparts including Bangladesh (4.4%), India (3.9%), Pakistan (3.0%), and Bhutan (3.7%) have a similar rate like Nepal.

The age-standardized prevalence of most of the mental disorders was not found identical by sex. Major depressive disorders and anxiety disorders were prevalent in females, whereas attention deficit, hyperactive disorder, conduct disorder, and autism spectrum disorders were manifested predominantly in males. Similar phenomena of sex-deferential distribution of mental disorders were observed in India and globally (Erskine et al., 2014; Rehm and Shield, 2019; Sagar et al., 2020). Some of the previous studies have indicated that the higher susceptibility of women to depressive and anxiety disorders could be linked to gender discrimination, gender-based violence, antenatal and postnatal stress, and adverse socio-cultural norms (Beydoun et al., 2012; Albert, 2015; Sagar et al., 2020). Likewise, a striking male bias in the prevalence of autism spectrum disorders and attention deficit hyperactive disorder could also be explained by sex-differential genetic and hormonal factors (Werling and Geschwind, 2013).

No evidence for an increased prevalence of overall mental disorders was found in Nepal between 1990 and 2019. Although the crude number of cases slightly increased both in males and females, the age-standardized prevalence of overall mental disorders decreased from 13.8% to 13.4% in the same period. The discrepancy between crude and age-standardized rates is mainly explained by population growth and changing age structures (Baxter et al., 2014). This finding is also opposite to the general

expectation that the prevalence of mental disorders might have significantly increased during the last two decades when psychological stressors including conflicts, natural disasters, and socio-economic adversities were abundant in Nepal. That might be because of the underreporting of mental disorders due to the stigma associated with mental illness in the community (Luitel et al., 2013; Devkota et al., 2021). The lack of trained health workers and accessible care might also have partly hindered the detection of cases, thereby causing an underestimation of mental disorders in Nepal (Upadhaya et al., 2017).

Unlike the prevalence, the DALYs for mental disorders marginally increased between 1990 and 2019. The DALYs due to mental disorders had a larger contribution to the total burden of disease, which has tripled in the last three decades. The multiplication of proportional contribution is caused by the decline in maternal and child mortality and deaths due to other communicable diseases. The proportional contribution of communicable maternal, neonatal, and nutritional diseases to total DALYs decreased from 70% to 29% between 1990 and 2019 (Institute of Health Metrics and Evaluation, 2019). The contribution of mental disorders to total disease burden (5.5% of all DALYs in 2019) in Nepal is comparable with that of India (4·7% of the total DALYs in 2017) (Sagar et al., 2020) and Mediterranean regions (4·7% of the total DALYs in 2015) (Charara et al., 2018). Most of the DALYs due to mental disorders are contributed by major depressive disorders, followed by anxiety disorders, idiopathic developmental intellectual

disability, and schizophrenia in Nepal and globally (Charara et al., 2018; Rehm and Shield, 2019; Sagar et al., 2020).

Over the past few years, several initiatives have been taken to improve mental health services in Nepal. Recently, the Ministry of Health and Population (MoHP) has developed a range of evidence-based training packages in mental health care for primary and community healthcare workers and has included six new psychotropic medicines in the list of free drugs (Luitel et al., 2020). Four priority mental and neurological disorders including depression, anxiety, psychosis, and epilepsy have also been included in the basic health care package (Ministry of Health and Population, 2018). Even though the government of Nepal has now given much emphasis to making mental health services available (i.e., addressing supply-side barriers) in the primary and community health care systems, evidence suggests that making mental health services available does not necessarily improve the help-seeking behaviors of people with mental health care. Effective implementation of mental health services in primary care has been challenging due to limited mental health awareness, low perceived need for mental health services, and high level of stigma in the wider Nepalese community which could negatively affect help-seeking and hence in early detection and management of people with mental health conditions (Luitel et al., 2020; Devkota et al., 2021). Therefore, community-level interventions should be developed for the promotion of mental health and prevention of mental disorders. The community-level intervention should target minimizing demand-side barriers which are considered as major barriers to mental health care (Luitel et al., 2020).

The major limitations of the current study are embedded with the GBD method of estimating the burden of mental disorders. Some argue that the current method of estimating mental disorders underestimates the burden of mental illness (Whiteford et al., 2016). The reason is due to the overlap between psychiatric and neurological disorders and excluding suicide and self-harm from the mental disorder category, among others (Whiteford et al., 2016). The GBD findings produced through modeling of a number of direct population data and covariates from Nepal could introduce biases in the estimates. However, it is important to note that this study provides the best possible estimates of the burden of mental disorders using the available data in Nepal. Likewise, this is also the first study to report DALYs due to mental disorders in Nepal. Our study findings are important in terms of illustrating the growing burden of mental disorders in Nepal and informing policymaking and program design in the Nepalese context. A clear understanding of the magnitude and distribution of the prevalence and burden of each mental disorder may help stakeholders tailor the psychosocial and mental health interventions specific to the disease, age, and sex in Nepal.

## Conclusions

The current study demonstrated that the proportional contribution of mental disorders to total DALYs is growing in Nepal. The burden of mental disorders largely varies by sex and age. To address the growing burden of mental disorders, there is a need for accelerating age, sex, and disease-specific promotive, preventive, and curative mental health interventions in Nepal.

**Open peer review.** To view the open peer review materials for this article, please visit http://doi.org/10.1017/gmh.2023.55.

**Supplementary material.** The supplementary material for this article can be found at http://doi.org/10.1017/gmh.2023.55.

**Data availability statement.** Data used in the study are publicly available on the official website of the Institute for Health Metrics and Evaluation.

**Acknowledgements.** We would like to thank the Institute for Health Metrics and Evaluation, University of Washington for allowing access to the data.

**Author contribution.** R.R.D. and A.R.P. conceptualized and interpreted the findings of the study. R.R.D. analyzed the data and prepared the first draft of the manuscript. A.R.P., S.J., N.L., K.M., K.A., M.D. interpreted the findings and reviewed the draft of the manuscript. All authors read and approved the final manuscript.

**Financial support.** The authors received no specific grant from any funding agency for this work.

**Competing interest.** The authors declare that the research was conducted in the absence of any commercial or financial relationships that could be construed as a potential conflict of interest.

**Ethics standard.** This study used the GBD data from the Institute for Health Metrics and Evaluation, University of Washington. No ethical approval is required.

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
