## [Reviewer Report]

Dear Editors,

We would like to submit our manuscript entitled “The Burden of Mental Disorders in Nepal between 1990-2019: Findings from the Global Burden of Disease Study 2019” to Cambridge Prisms: Global Mental Health for your consideration.

Mental disorders are the major cause of disease burden globally. Despite having a dire situation of mental health system in Nepal (which is highlighted by our co-authors elsewhere[1-4]), little is known about the burden of mental disorders in Nepal.

In the current study, we thoroughly investigated and quantified the burden of mental disorders across different age groups and sex in Nepal using the Global Burden of Disease Study 2019. We found that the contribution of mental disorders to total disease burden has tripled between 1990 and 2019 in Nepal. Our study findings are important in terms of illustrating the growing burden of mental disorders in Nepal and informing policymaking and programme designs in the Nepalese context.

We would be very grateful if you would consider the manuscript for publication in your journal.

Yours sincerely,

Dr Raja Ram Dhungana

Kathmandu, Nepal

References

1. Luitel NP, Jordans MJ, Adhikari A, Upadhaya N, Hanlon C, Lund C, et al. Mental health care in Nepal: current situation and challenges for development of a district mental health care plan. Confl Health. 2015;9:3. Epub 2015/02/20. doi: 10.1186/s13031-014-0030-5. PubMed PMID: 25694792; PubMed Central PMCID: PMCPMC4331482.

2. Luitel NP, Garman EC, Jordans MJD, Lund C. Change in treatment coverage and barriers to mental health care among adults with depression and alcohol use disorder: a repeat cross sectional community survey in Nepal. BMC Public Health. 2019;19(1):1350. Epub 2019/10/24. doi: 10.1186/s12889-019-7663-7. PubMed PMID: 31640647; PubMed Central PMCID: PMCPMC6806507.

3. Luitel NP, Jordans MJD, Kohrt BA, Rathod SD, Komproe IH. Treatment gap and barriers for mental health care: A cross-sectional community survey in Nepal. PLoS One. 2017;12(8):e0183223. Epub 2017/08/18. doi: 10.1371/journal.pone.0183223. PubMed PMID: 28817734; PubMed Central PMCID: PMCPMC5560728.

4. Dhungana RR, Aryal N, Adhikary P, Kc RK, Regmi PR, Devkota B, et al. Psychological morbidity in Nepali cross-border migrants in India: a community based cross-sectional study. BMC Public Health. 2019;19(1):1534. doi: 10.1186/s12889-019-7881-z.

---

## [Reviewer Report]

The manuscript is well written. This particular study has assessed the prevalence and disability-adjusted life-years (DALY) of mental disorders in Nepal between 1990 and 2019 using the Global Burden of Disease Study 2019. The authors concluded that the proportional contribution of mental disorders to the total disease burden has increased significantly in the last three decades in Nepal which is a crucial and important finding. However, the following concerns are needed to be addressed

Major Comments

1. The method section of the paper should be more detailed. The authors should mention the study design and steps of data extraction, inclusion, and exclusion criteria clearly (for example PRISMA). Although it is mentioned “A detailed description of the metrics, data sources, and statistical modeling for mental disorders have been reported elsewhere (Vos et al. 2020)” at least a minimum description is required to be incorporated in the method section of the manuscript for clarity.

2. Please add how you define DALY and mental disorder for this particular paper in simple sentences in the method section, can add a subheading “study outcome” if possible. “Mental disorder” is a general term.

Minor comments:

1. Impact “Statement” Spelling needs to be corrected ( page 1, before abstract).

2. The study design should be mentioned in the abstract.

---

## [Reviewer Report]

Dear authors,

You have highlighted very well the prevalence of common mental illnesses and the disability associated with the diseases based on the findings of the Global Burden of Disease (GBD) Study 2019. In the context of Nepal’s developing mental health policy and planning in recent years, this study will be extremely important to the relevant stakeholders in promoting mental health in Nepal. The only caveat is that the finding has to depend on the secondary data from the GBD study. I found the manuscript very well written and will be a great asset in the field of global mental health.

---

## [Reviewer Report]

Dear Editor,

We would appreciate for your and reviewers’ time and thoughtful review of our manuscript titled " The Burden of Mental Disorders in Nepal between 1990-2019: Findings from the Global Burden of Disease Study 2019”. Insightful comments and constructive feedback from Editor and reviewers were invaluable in refining the quality and depth of our work.

We have carefully addressed each of comments and suggestions made by Editor and reviewers, and we are pleased to provide a detailed response outlining the revisions we have made to the manuscript. Your feedback has guided us in enhancing the clarity, methodology, and overall coherence of the paper.

Below, we summarize the key changes we have implemented based on your feedback:

-Structured the manuscript based on journal guidelines,

-Included graphical abstract,

-Revised methods section,

-Provided point to point responses to the reviewers’ comment

Regards,

Dr Raja Ram Dhungana

Corresponding Authors

raja.dhungana@gmail.com

Kathmandu, Nepal

---

## [Reviewer Report]

Dear authors,

I appreciate your efforts to bring these important findings related to the burden of mental illnesses in Nepal. Your manuscript is very well written. I like the discussion part where you have tried to address some of the limitations of the study. Your article will be an invaluable asset to the literature related to mental health in LMICs.